# *Zat12* Gene Ameliorates Temperature Stress in Wheat Transgenics by Modulating the Antioxidant Defense System

**Manpreet Kaur [1], Bavita Asthir [1,*], Ramandeep Kaur [2] and Ankur Chaudhary [3]**

1 Department of Biochemistry, Punjab Agricultural University, Ludhiana 141004, Punjab, India
2 School of Agricultural Biotechnology, Punjab Agricultural University, Ludhiana 141004, Punjab, India
3 Department of Agronomy, CCS Haryana Agricultural University, Hisar 125004, Haryana, India
* Correspondence: b.asthir@rediffmail.com or basthir@gmail.com

**Abstract:** The present study was undertaken with the objective to reconnoiter the role of *Zat12*-related biochemical activities in temperature stress tolerance in wheat transgenic lines Z-8-12 1A, Z-8-12 1B, Z-8-19, and Z-15-10, which were produced by transforming wheat-cultivar PBW 621. *Zat12* transgenics (ZT) along with non-transgenic (NT) wheat cultivars (PBW 621, PBW, 550, and HD 3086) were assessed at the three-weeks seedling stage under chilling ($-2\,^{\circ}$C and $-4\,^{\circ}$C) and heat ($30\,^{\circ}$C and $32\,^{\circ}$C) stress. Specific activities of superoxide dismutase (SOD), peroxidase (POD), ascorbate peroxidase (APX), glutathione-S-transferase (GST), glutathione reductase (GR), and antioxidants (proline and ascorbate) were profoundly increased under temperature stress in ZT related to NT. However, under $-4\,^{\circ}$C and $32\,^{\circ}$C, a significantly higher increase was reported. In contrast, $H_2O_2$ and MDA were found to be much lower in ZT than in NT. Similarly, lesser decreases in length, fresh weight, and dry weight of seedlings were reported in ZT at $30\,^{\circ}$C and $32\,^{\circ}$C. RT-PCR studies revealed the enhanced expression of *Zat12* in the roots of seedlings at the 5, 10, and 14 days after germination (DAG) stages in ZT under the stress conditions. Upregulation of the antioxidant defense system in ZT and their better tolerance depict an alternative for wheat cultivation under temperature stress-prone areas.

**Keywords:** antioxidative enzymes; ascorbate; proline; RT-PCR; transformed

## 1. Introduction

Wheat (*Triticum aestivum* L.) is a significant primary food crop grown worldwide, with a yearly production of about 761.5 million tonnes [1]. Its contribution is nearly 20% of dietary proteins and calories, all-inclusive [2]. However, changes in climate have intensified the severity of numerous abiotic stresses including high temperature, drought, salinity, flooding, and cold stress, which significantly reduce major crop plants' yield [3]. Recently, the present levels of wheat productivity became impotent to cope with the demand of the growing population, which may lead to price uncertainty and hunger uprisings, and is the main challenge for breeders globally. In wheat, high temperature is among the most vital environmental factors limiting its yield and productivity. Every $1\,^{\circ}$C or $2\,^{\circ}$C rise above the mean temperature ($23\,^{\circ}$C) declines its yield by from $-2.3$% to 7.0% and $-2.4$% to 10.5%, respectively [4]. High temperature distresses the yield either by persistent and continuous exposure to high temperature until $32\,^{\circ}$C, or else by abrupt heat shock with transitory exposure to $33\,^{\circ}$C or above. Moreover, in many areas, wheat crops suffer a freezing injury of fluctuating sternness, causing significant yield losses [5]. Winter wheat is more likely to experience extreme subzero temperatures of both air and soil during winters, which affect its yield, and winter wheat requires the tolerance mechanism of cold acclimation to survive under low temperatures. To meet the augmented demand by 2050, the production of wheat must be raised by 60%, which sequentially necessitates its tolerance to various abiotic stresses.

Complex networks of regulatory genes that constitute distinctive stress-responsive regulons impart significant role acclimatization to environmental disturbances. A zinc-finger transcription protein has been encoded by *Zat12* [6] and is one of the vital factors which reacts and responds to a large number of abiotic stresses and imparts an imperious role in oxidative stress management in *Arabidopsis thaliana* [7,8]. The *Zat12* gene responds to several abiotic stress conditions viz. heat, cold [9], heavy metals [10], light [11], and low oxygen stresses [12]. Reactive oxygen species (ROS) accumulate in the cytosol and are recognized by several transcription factors which respond to stress conditions such as *Zat* family members (Zat12, 10, and 7) [13,14], and then activate the members of other transcription coactivators which modulate the defense process and act as ROS scavengers [15]. Vogel et al. [16] reported that COS (Cold Standard Set) genes are responsive to *Zat12* overexpression in *Arabidopsis*. Chandra et al. [17] demonstrated that *Zat12* tomato transformants formed from *Brassica carinata* conferred tolerance to abiotic stress conditions. Therefore, this stress tolerance behavior makes *Zat12* an attractive subject for biochemical study in cereals, especially in wheat crops.

The advancement of stress-tolerant crop cultivars is one of the foremost concerns in breeding programs. The tolerance of plants can be enhanced in several ways, one of which includes the use of transgenics. In the context of the above, to make the wheat crop tolerant to temperature stress, *Zat12*-transformed wheat lines were produced against PBW621, and it was found that they probably be a good alternative to meet the growing demand. In this study, we focused on four *Zat12* wheat transgenic lines viz. Z-8-12 1A, Z-8-12 1B, Z-8-19, and Z-15-10 together with recipient PBW621, which is a wheat cultivar of the North Western Plains of India. As well, two other wheat cultivars, i.e., HD3086 and PBW550, were also included for comparison. This study evaluated various physio-biochemical traits viz. antioxidative enzymes, antioxidants, osmolytes, lipid peroxidation, root length, fresh weight (FW), and dry weight (DW) in *Zat12* transgenics and non-transformed cultivars of wheat at the early stages of seedlings under heat and cold stress to examine how the antioxidant defense system responds under different stress conditions. Furthermore, the expression analysis of the *Zat12* gene through RT-PCR under non-stress and stress conditions in wheat transgenics was done. Hence, this study demarcates the biochemical consequences of stress-tolerant wheat transgenics having constitutively expressed *Zat12*, which regulates metabolic steps of the antioxidant defense system.

## 2. Results

### 2.1. Biochemical Analysis

The higher specific activities APX, SOD, POD, GST, and GR in roots of three-week-old seedlings of ZT lines viz, Z-8-12 1A, Z-8-12 1B, Z-8-19, and Z-15-10 as compared to NT wheat cultivars (PBW621, PBW550, and HD3086) under high temperature and cold stress were observed in this study.

A steady rise in the specific activity of APX was detected in transgenics under temperature stress. This value was in the range of 49.21–59.35 µmoles of MDA produced per $min^{-1}g^{-1}$ FW under cold stress (66.02–74.97% increase) and 40.21–47.07 µmoles of MDA produced per $min^{-1}g^{-1}$ FW under heat stress in ZT lines whereas, in the case of NT cultivars, it was in the range of 22.4–35.7 µmoles of MDA formed per $min^{-1}g^{-1}$ FW (Figure 1). The SOD-specific activity increased to 3.21–3.76 Units per $min^{-1}g^{-1}$ FW in ZT lines (35.02–51.61% increase) whereas, under controlled conditions, it was in the range of 1.85–2.20 Units per $min^{-1}g^{-1}$ FW (16.35–38.36% increase). Higher SOD-specific activity was observed at −4 °C than at −2 °C in all of the ZT lines, with more at 32 °C than at 30 °C. Z-8-12 1B and Z-8-12 1A showed a maximum increase of 51.61% and 47.58%, respectively, in SOD activity at −4 °C (Figure 1). The specific activity of POD was also found to be increased under temperature stress in ZT lines; however, it was increased comparatively less than APX activity, which showed more than a 2.6-fold increase compared to PBW621. At −4 °C cold stress, the POD activity was in the range of PBW 621 (31.2 ΔO.D $min^{-1}g^{-1}$ FW) < Z-8-19 (46.6 ΔO.D $min^{-1}g^{-1}$ FW) (49.35% increase) < Z-8-12 1A (47.1 ΔO.D $min^{-1}g^{-1}$

FW) (50.96% increase) < Z-15-10 (47.8 ΔO.D min$^{-1}$g$^{-1}$ FW) (53.20% increase) and Z-8-12 1B (48.8 ΔO.D min$^{-1}$g$^{-1}$ FW) (56.41% increase) (Figure 1).

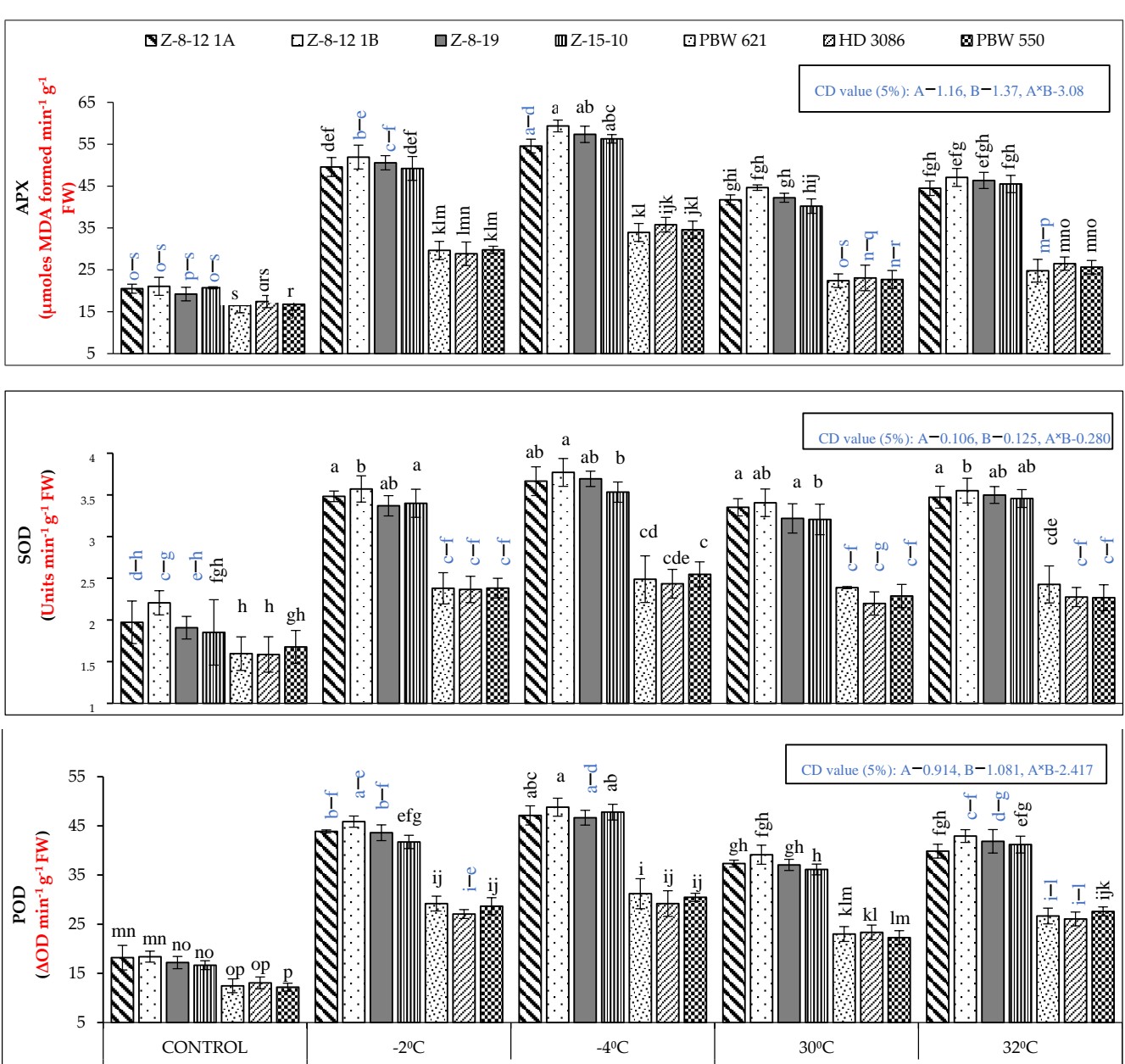

**Figure 1.** Antioxidant enzymes ascorbate peroxidase (APX), superoxide dismutase (SOD), and peroxidase (POD) activities in the root of *Zat12* transgenic lines and non-transformed (PBW 621, HD 3086, and PBW 550) wheat genotypes at 21 days after germination (DAG) under different temperatures. Data represent the mean of triplicates and error bars signify standard deviations of triplicates. Data were evaluated by two-way ANOVA in accordance with values that are significant at a 5% level of significance, where A, B, and AB represent CD values among wheat transgenics/cultivars, between temperatures, and their interaction, respectively. Different letters elucidate the significant differences obtained by Tukey's post hoc test amongst the wheat transgenics/cultivars and treatments ($p < 0.05$).

Similarly, the GR-specific activity increased in the ZT lines as well as in the NT wheat cultivars under temperature stress. However, the rise was meaningfully higher in transgenics as compared to NT genotypes. All-out GR activity was noted in Z-8-12 1B (72.47% increase) > Z-8-12 1A (70.57% increase) > Z-8-19 (69.45% increase) > Z-8-19 (68.23%

increase) at −4 °C (Figure 2). The GST-specific activity was increased significantly (nearly about 2.5-fold) in the ZT lines under temperature stress than in the NT wheat cultivars. Heat stress at 32 °C and cold stress at −4 °C showed the maximum increase in GST activity in *Zat12* wheat transgenics, whereas the maximum activity was recorded at −4 °C in Z-8-12 1B (89.07% increase) followed by Z-8-19 (85.53% increase), Z-8-12 1A (84.92% increase), and Z-15-10 (78.30% increase), as compared to PBW621.

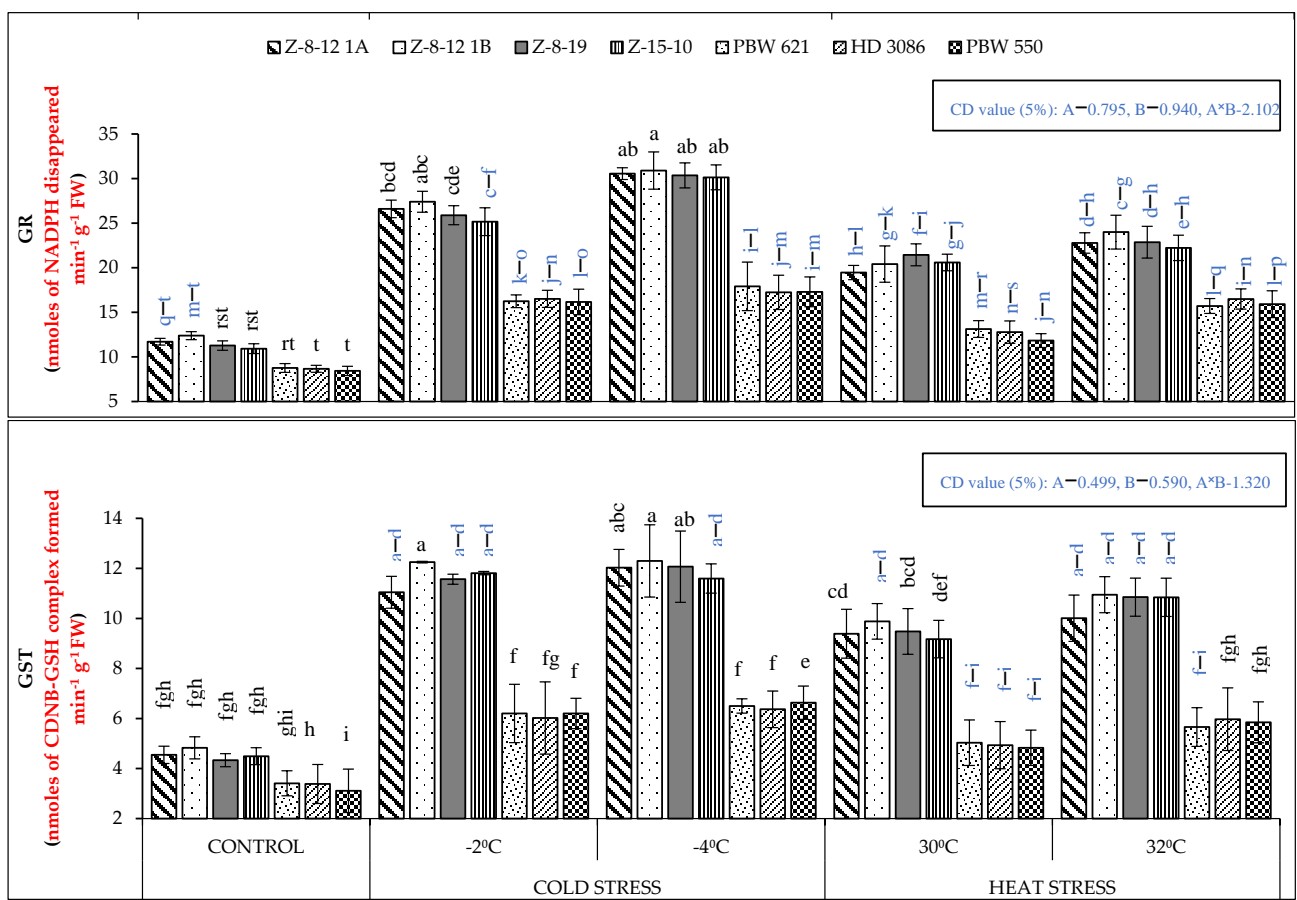

**Figure 2.** Antioxidant enzymes glutathione reductase (GR) and glutathione-s-transferase (GST)'s activities in the root of *Zat12* transgenic lines and non-transformed (PBW 621, HD 3086, and PBW 550) wheat genotypes at 21 days after germination (DAG) under different temperatures. Data represent the mean of triplicates and error bars signify standard deviations in triplicates. Data were evaluated by two-way ANOVA in accordance with values that are significant at a 5% level of significance, where A, B, and AB represent CD values among wheat transgenics/cultivars, between temperatures, and their interaction, respectively. Different letters elucidate the significant differences obtained by Tukey's post hoc test amongst the wheat transgenics/cultivars and treatments ($p < 0.05$).

The *Zat12*-transformed wheat lines, although only one (Z-8-19) showed at least more than a two-fold upsurge in the content of proline under stress conditions. Z-8-12 1B (92.41% increase) and then Z-8-12 1A (86.20% increase) showed a higher increase in proline content than Z-15-10 (85.51% increase) and Z-8-19 (79.31% increase), in contrast to PBW621 at −4 °C (Figure 3). Proline content was higher in the seedlings at 32 °C than at 30 °C in all of the transformed wheat lines (Figure 3). Temperature stress increased the ascorbate content in the roots of the studied ZT lines (36% to 65%) in NT wheat cultivars (19% to 29%) as compared to control conditions (Figure 3). The uppermost ascorbate accumulation was detected at −4 °C and 32 °C in Z-8-12 1B (67%), followed by Z-8-12 1A (65%). The $H_2O_2$ content increased in transgenics less than 2-fold under temperature stress, while the supreme increase was noted in PBW621, which is more than 2.8-fold, and other non-

transformed wheat cultivars. At −4 °C, the amount was the lowest in Z-8-12 1B and the highest in PBW621 (Figure 3). Accordingly, the ZT lines showed significantly less $H_2O_2$ content than the NT genotypes, consequently diminishing the toxicity. The level of malondialdehyde in ZT lines under the stress laid in the range 5.2–6.8 µmole g$^{-1}$ FW, which is near about a two-fold increase from the controlled conditions (Figure 3). However, in the case of the non-transformed lines, i.e., PBW 621, HD 3086, and PBW 550, it laid in the range 9.17–11.54 µmole g$^{-1}$ FW, which was near about a four-fold increase in malondialdehyde content under stress. Table 1 shows the ANOVA table for biochemical parameters in wheat transgenics/cultivars grown under different temperatures.

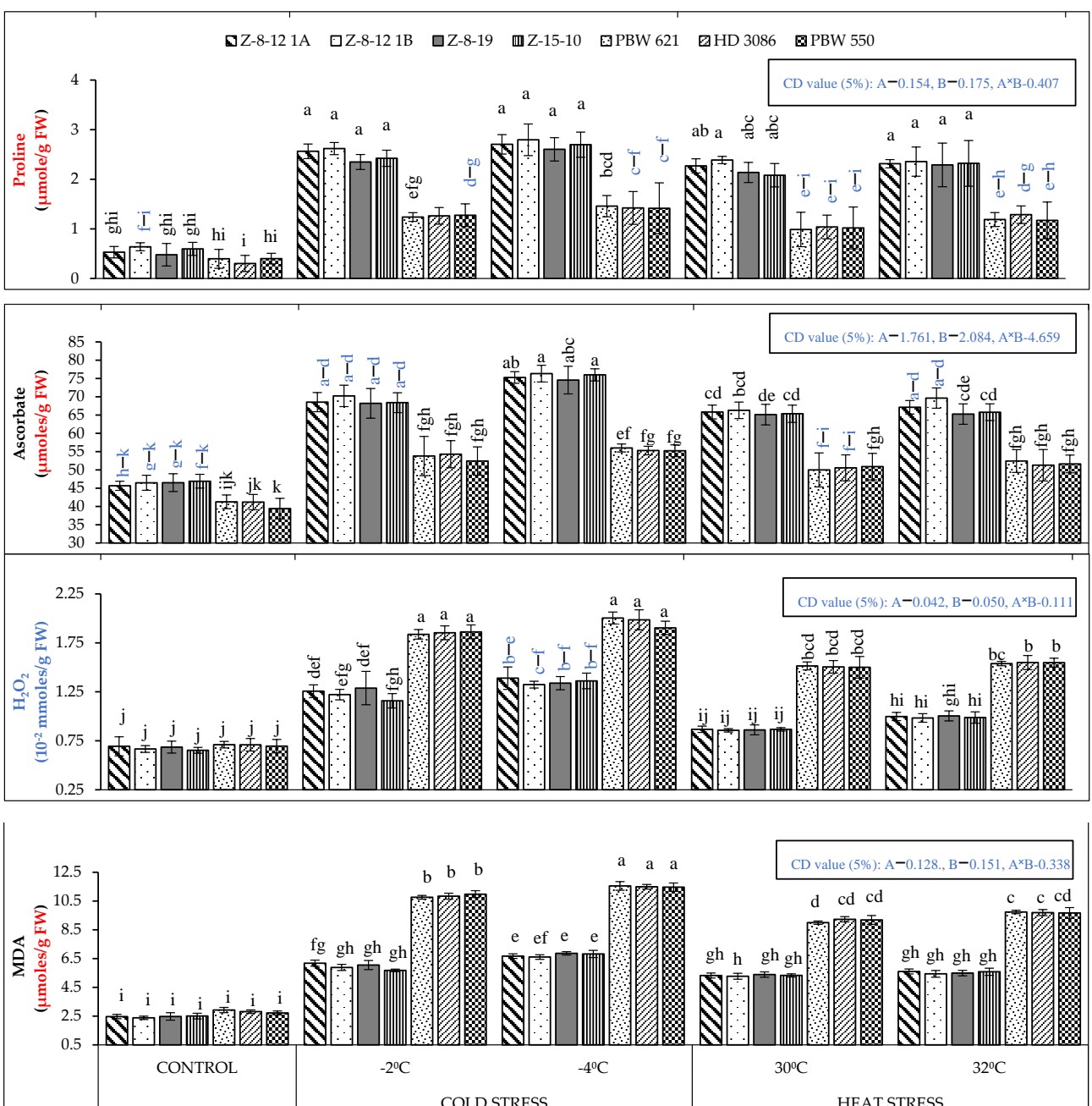

**Figure 3.** Content of proline, ascorbate, $H_2O_2$, and malondialdehyde (MDA) in the root of *Zat12* transgenic lines and non-transformed (PBW 621, HD 3086, and PBW 550) wheat genotypes at 21DAG under different temperatures. Data represent the mean of triplicates and error bars signify standard deviations in triplicates. Data were evaluated by two-way ANOVA in accordance with values that are

significant at a 5% level of significance, where A, B, and AB represent CD values among wheat transgenics/cultivars, between temperatures, and their interaction, respectively. Different letters elucidate the significant differences obtained by Tukey's post hoc test amongst the wheat transgenics/cultivars and treatments ($p < 0.05$).

**Table 1.** Effect of different temperatures on biochemical parameters in wheat transgenics/cultivars.

| Treatment | APX | SOD | POD | GR | GST | PRO | ASC | H$_2$O$_2$ | MDA |
|---|---|---|---|---|---|---|---|---|---|
| Factor A (Temperature) | | | | | | | | | |
| 20 °C | 18.882 | 3.104 | 35.810 | 21.518 | 9.156 | 1.948 | 62.732 | 1.013 | 4.864 |
| −2 °C | 41.361 | 3.273 | 39.343 | 23.488 | 10.081 | 2.158 | 66.667 | 1.048 | 5.393 |
| −4 °C | 47.402 | 2.935 | 33.981 | 20.137 | 8.882 | 1.828 | 61.278 | 0.963 | 4.891 |
| 30 °C | 33.852 | 2.297 | 25.629 | 15.025 | 5.595 | 1.123 | 51.852 | 1.635 | 9.407 |
| 32 °C | 37.182 | 2.234 | 24.371 | 14.083 | 5.360 | 1.087 | 50.201 | 1.509 | 8.953 |
| CD (5%) | 1.166 | 0.106 | 0.914 | 0.795 | 0.499 | 0.154 | 1.761 | 0.042 | 0.128 |
| Factor B (Lines/Cultivar) | | | | | | | | | |
| Z-8-12 1A | 42.165 | 2.764 | 32.280 | 18.889 | 7.767 | 1.576 | 58.321 | 1.273 | 6.882 |
| Z-8-12 1B | 44.808 | 2.702 | 31.387 | 18.501 | 7.577 | 1.658 | 57.801 | 1.264 | 6.998 |
| Z-8-19 | 43.137 | 2.937 | 33.400 | 20.185 | 8.585 | 1.773 | 60.540 | 1.148 | 5.882 |
| Z-15-10 | 42.379 | 2.721 | 31.533 | 18.581 | 7.435 | 1.582 | 58.636 | 1.263 | 7.012 |
| PBW 621 | 25.452 | 2.837 | 32.627 | 18.995 | 8.151 | 1.693 | 59.419 | 1.153 | 6.183 |
| HD 3086 | 26.323 | 2.802 | 31.120 | 18.620 | 7.718 | 1.570 | 58.395 | 1.269 | 6.932 |
| PBW 550 | 25.894 | 2.681 | 30.440 | 18.180 | 7.472 | 1.549 | 56.708 | 1.265 | 7.022 |
| CD (5%) | 1.379 | 0.125 | 1.081 | 0.940 | 0.590 | 0.175 | 2.084 | 0.050 | 0.151 |

APX: Ascorbate Peroxidase; SOD: Superoxide Dismutase; POD: Peroxidase; GR: Glutathione Reductase; GST: Glutathione-S-Transferase; PRO: Proline; ASC: Ascorbate; H$_2$O$_2$: Hydrogen Peroxide; MDA: Malondialdehyde.

### 2.2. Molecular Analysis

The PCR-based assay clearly depicted a *Zat12* gene presence in transgenics. A product (812 bp) was inveterate by comparing it with a 1 kb DNA ladder as standard. Further confirmation of the *Zat12* gene was revealed by using the CaMV 35S promoter sequence in the forward direction and the sequence of *Zat12* in the reverse direction. The expression of *Zat12* was analyzed by RT-PCR transgenics seedlings. A very low level of expression was detected at 5DAG in comparison to 10DAG in transgenics, and the highest level of expression was observed under the stress conditions −2 °C and 32 °C (Figure 4).

### 2.3. Growth Parameter Analysis

The length, FW, and DW were meaningfully reduced with temperature stress; however, there was less reduction in the case of *Zat12* transgenics than in the non-transformed PBW621. With the exposure to high-temperature stress of 30 °C and 32 °C, shoot length was abridged by 23% and 29%, root length by 13% and 21%, shoot FW by 13% and 20%, root FW by 5% and 6.5%, shoot DW by 9% and 15%, and root DW by 5% and 9%, respectively, when compared with control (Figures 5 and 6). Z-812 1B and Z-8-12 1A were less exaggerated under high temperatures, whereas PBW621 was affected the most. The supreme reduction in root length was detected in PBW621 (40%), although the smallest decrease was observed in *Zat12* transgenic Z-8-12 1B (13%). PBW621 presented the highest decrease in shoot fresh and dry weight, i.e., 33% and 36% at 32 °C. Table 2 shows the ANOVA table for growth parameters in the wheat transgenics/cultivars grown under heat stress.

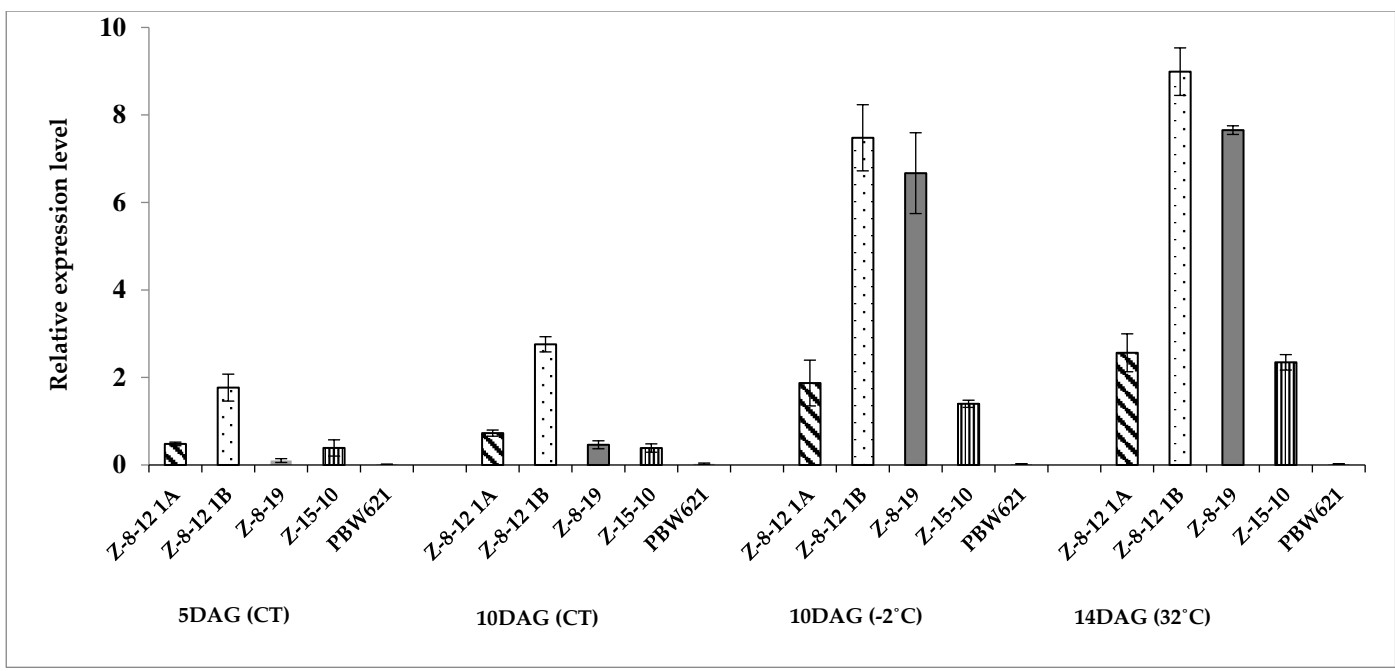

**Figure 4.** Relative expression of *Zat12* gene at different seedling stages, i.e., 10 days after germination (DAG) and 14 days after germination (DAG) under control (25 °C) (CT), cold (−2 °C), and heat stress (32 °C) in root tissue of various *Zat12* wheat transgenics and NT (PBW621) wheat cultivar, in the background of which transgenics were made.

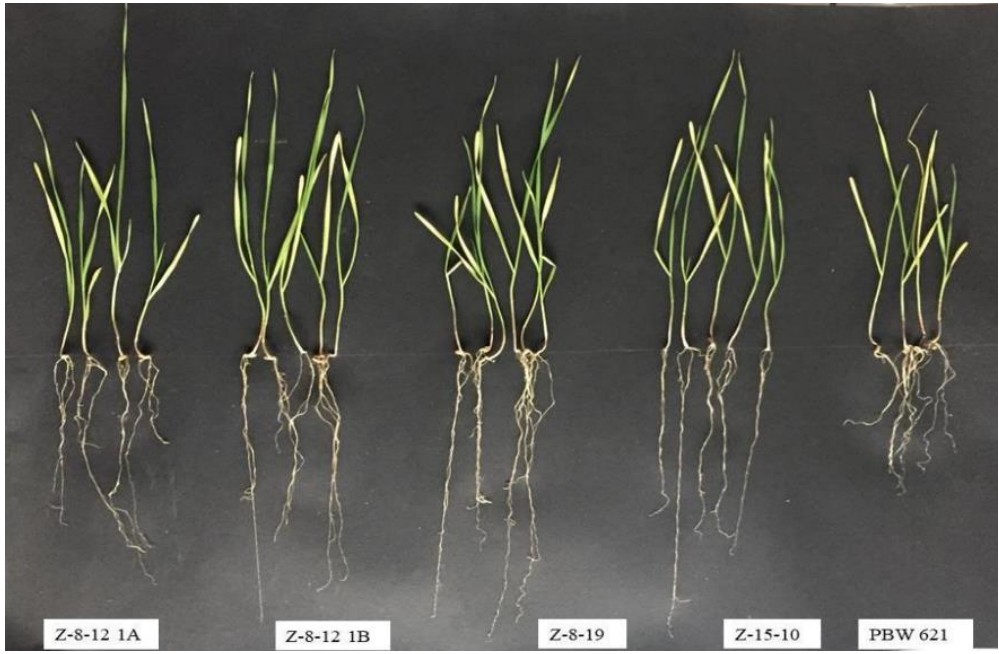

**Figure 5.** The seedlings of *Zat12*-transformed wheat lines and non-transformed wheat genotype PBW621 grown under the heat stress (32 °C) at 15 DAG stage.

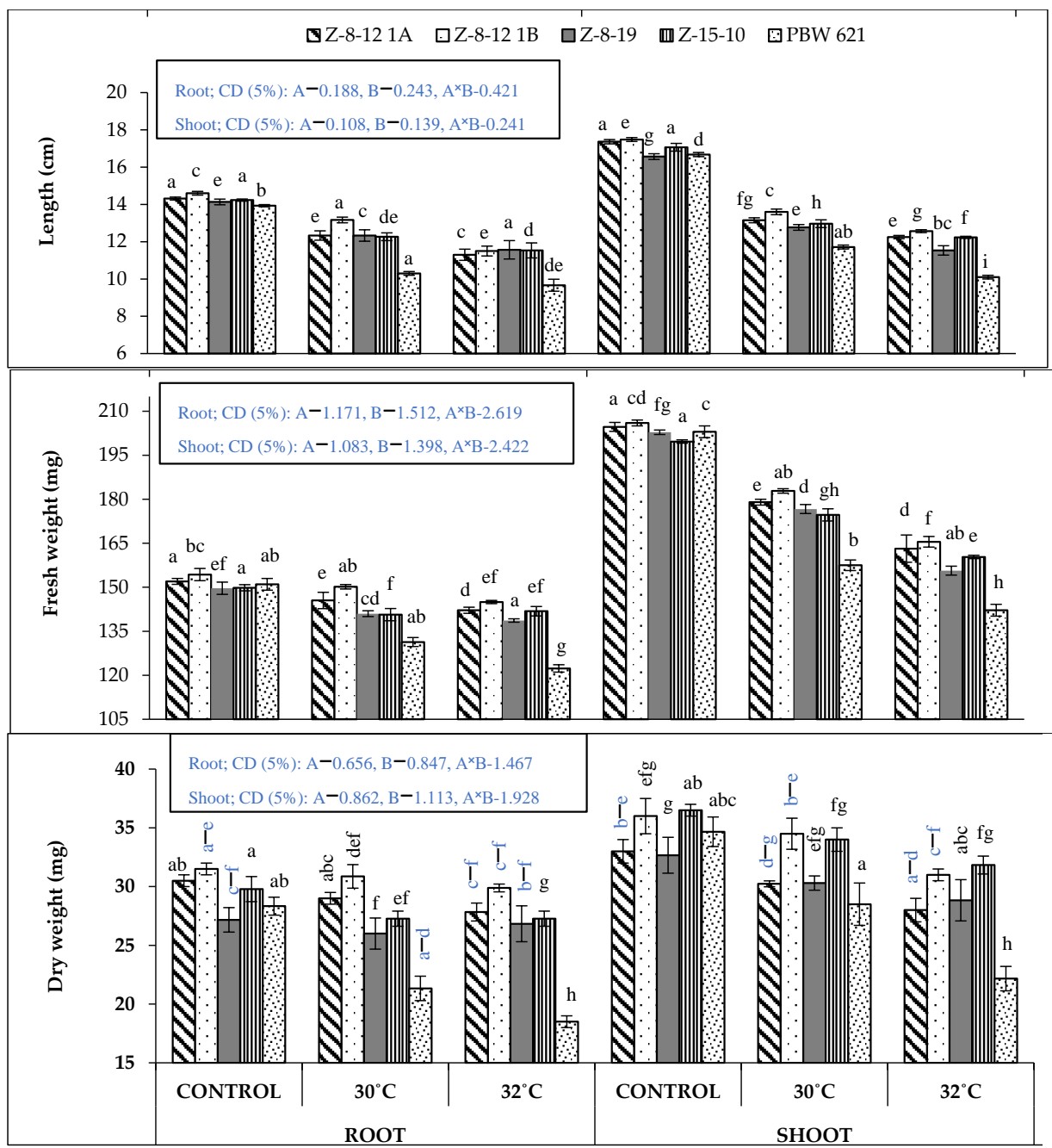

**Figure 6.** Heat stress-induced changes in the length (L), fresh weight (FW), and dry weight (DW) of root and shoot at 15 days after the germination (DAG) stage under different temperatures. Values are the means of triplicates and error bars signify standard deviations of triplicates. Data were evaluated by two-way ANOVA in accordance with values that are significant at a 5% level of significance, where A, B, and AB represent CD values among wheat transgenics/cultivars, between temperatures, and their interaction, respectively. Different letters elucidate the significant differences obtained by Tukey's post hoc test amongst the wheat transgenics/cultivars and treatments ($p < 0.05$).

**Table 2.** Effect of heat stress on growth parameters in wheat transgenics/cultivars.

| Treatment | RL | SL | RFW | SFW | RDW | SDW |
|---|---|---|---|---|---|---|
| Factor A (Temperature) | | | | | | |
| 20 °C | 12.343 | 14.570 | 146.833 | 184.467 | 29.940 | 32.347 |
| 30 °C | 11.753 | 13.700 | 140.827 | 176.067 | 27.933 | 31.860 |
| 32 °C | 10.940 | 12.539 | 135.433 | 165.533 | 24.540 | 30.233 |
| CD (5%) | 0.188 | 0.108 | 1.171 | 1.083 | 0.656 | 0.862 |
| Factor B (Lines/Cultivar) | | | | | | |
| Z-8-12 1A | 11.694 | 13.967 | 142.544 | 178.278 | 29.217 | 32.667 |
| Z-8-12 1B | 11.667 | 13.650 | 142.333 | 177.389 | 27.811 | 31.578 |
| Z-8-19 | 11.522 | 13.564 | 141.389 | 176.500 | 27.389 | 30.989 |
| Z-15-10 | 11.489 | 13.244 | 138.111 | 169.722 | 26.556 | 31.111 |
| PBW 621 | 12.022 | 13.589 | 140.778 | 174.889 | 26.383 | 31.056 |
| CD (5%) | 0.243 | 0.139 | 1.512 | 1.398 | 0.847 | 1.113 |

RL: Root Length; SL: Shoot Length; RFW: Root Fresh Weight; SFW: Shoot Fresh Weight; RDW: Root Dry Weight; SDW: Shoot Dry Weight.

## 3. Discussion

Abiotic stress leads to various biochemical and morphophysiological changes in plants, which results in a lethal effect on crop plants [18]. Temperature stress hastens ROS production together with singlet oxygen, superoxide radical, $H_2O_2$, and hydroxyl radical, thus persuading oxidative stress [19]. Antioxidant enzymes and growth parameters are dynamic for assessing the stability and cell homeostasis in contradiction of any type of stress. Transgenic plants, nowadays, outshine conventionally cultivated ones because of helpful genes inserted in the genome of the host [20]. This study was primarily undertaken to recognize the *Zat12*-induced biochemical responses, providing a foundation for additional understanding of the role of *Zat12* in temperature stress amelioration in wheat transgenics.

The specific activities of various antioxidative enzymes, including APX, SOD, POD, GST, and GR, increased under temperature stress conditions. Similar results were reported by Farooq et al. [21], that upon acquaintance with high temperature, especially during the reproductive phase, the antioxidant enzymes activities were significantly increased in wheat genotypes which are heat-tolerant. APX was observed to be an effective ROS regulator which catalyzes the detoxification of hydrogen and prevents cellular damage [22]. SOD and APX, along with POD, were observed to play a role as effective ROS quenchers and also prevent the peroxidation of lipids [23]. SOD is a metalloenzyme that is directly related to stress, and acts as the first defense and primary protector in plants against the oxidative stress produced by ROS. SOD acts on superoxide anions ($O_2^-$) and convert them into hydrogen peroxide ($H_2O_2$) and molecular oxygen, and, ultimately, reduce the superoxide anion level in plant cells, which is harmful at a high concentration [24]. The produced $H_2O_2$ is detrimental to the cells at excessive concentrations. APX has a robust affinity for $H_2O_2$, signifying that the ascorbate-glutathione cycle plays a vigorous role in ROS controlling. APX acts on $H_2O_2$ and forms water and oxygen by using ascorbate as a reducing molecule [25]. POD acts on $H_2O_2$ or further active organic hydroperoxides viz. lipid peroxides, and converts them into less reactive and stable species. The present results showed that the percent increase in a specific activity is much higher in APX than SOD, and POD under temperature stress (Figure 1). SOD showed modest activity in ZT lines; nevertheless, the level of POD increased under stress conditions. The hydrogen peroxide formed by SOD acts as a signaling molecule to upregulate the POD enzyme to convert it into less reactive and non-toxic molecules. In our results, a significantly higher APX activity (~70%) in wheat transgenics under temperature stress was reported. All wheat transgenic lines (Z-8-12 1A, Z-8-12 1B, Z-8-19, and Z-15-10) showed increased APX-specific activity, as it acts on hydrogen peroxide and reduces it into water and dehydroascorbic acid by using the reducing molecule ascorbate. As the seedlings were under temperature

stress, the oxidative stress increased correspondingly, so as to quench the accumulated superoxide free radical, while SOD-specific activity remained on the higher side in wheat transgenics. The higher specific activity of SOD in wheat transgenics could be responsible for the stimulation of the antioxidant defense system to remove superoxide ions and form hydrogen peroxide and more stable oxygen molecules. Similar results have been reported by Bozca and Leblebici [26], that APX activity increased in the roots of *Helianthus annuus* when grown at 15 °C. The present study depicted that excess $H_2O_2$ formed by SOD is possibly detoxified by the APX and POD enzyme, while it is majorly detoxified by APX, and helps in ameliorating the heat and cold stress in *Zat12* wheat transgenics. Likewise, it was previously reported that the APX enzyme progresses the tolerance to abiotic stress in transgenic *Arabidopsis thaliana* by reducing the accumulation of $H_2O_2$ [27].

Moreover, the other antioxidant enzymes GR and GST showed higher activity in transgenics, which might also be responsible for maintaining ROS homeostasis under heat and cold stress in seedlings. GR acts as a key component in the ascorbate-glutathione pathway by catalyzing the rate-limiting or last step of this pathway to maintain a higher ratio of GSH/GSSG [28]. In our study, GR activities increased under temperature stress (both heat and cold) (Figure 2). A higher GR activity in the case of *Zat12*-transformed lines as compared to PBW621 indicated their vital role in maintaining the redox state of the cells. This indicates the better ability of the transgenics to combat temperature stress, which possibly is an outcome of gene expression variations and regulation of their synthesis. High GR activity helps in maintaining the reducing environment in the cells through coupling NADPH oxidation to glutathione disulfide reduction, producing reduced glutathione. It has been reported in the reviewed literature that GR activity increased in several plant species under different abiotic stresses, including salinity, heat, and chilling stress [29]. Moreover, GR activities are augmented in various stress-tolerant plants [30]. GST is an enzyme known for its ability to catalyze the conjugation of glutathione (GSH) to hydrophobic substrates and form less toxic peptide derivatives [31]. In our study, the activity of GST increased with increasing temperature; however, the increase was by the most in *Zat12* transgenics and by the least in PBW621 (Figure 2). Gao et al. [32] have reported that the rise in specific activities of GST deliberately improves antioxidative defense by scavenging ROS under salt stress in Poplar; therefore, it could act as a criterion for stress acclimatization. Higher activity of antioxidant enzymes in wheat transgenics would prevent the accumulation of ROS under temperature stress and helps in amelioration under such unfavorable conditions.

The other explanations could be ascribed to a significantly higher concentration ($p < 0.05$) of osmolytes in wheat transgenics. Proline accumulation is an inclusive plant's response to various stresses [33]. Proline is an osmolyte that acts as a ROS scavenger and a molecular chaperone that accumulates and helps in protection from environmental stress. Therefore, the high proline content in transformed wheat lines under heat and cold stress conditions (Figure 3) is correlated with improved tolerance to stress conditions in *Zat12* transgenics. Ascorbic acid, as an antioxidant, has considerable potential to not only scavenge ROS; nevertheless, it also cooperates with other antioxidants, together with tocopherol, thioredoxin, and glutathione. Moreover, it can also augment biosynthesis and antioxidant enzymes (SOD, CAT, and POD) [34]. The significantly higher ascorbate content in ZT as compared to NT wheat cultivars under temperature stress plays a vital role in transgenics to combat the stress conditions. It has been reported previously that ascorbate is highly effective in extenuating the distressing effects caused by chilling stress by depressing electrolyte leakage, and the peroxidation of lipids [35].

A higher concentration of $H_2O_2$ damages growth and development in plants by damaging macromolecules, especially membrane lipids [36], and is measured as an oxidative stress indicator [37]. Temperature stress raised the concentration of $H_2O_2$ in NT wheat cultivars (Figure 3). The $H_2O_2$ concentrations were significantly lowered in transgenics, suggesting a balanced $H_2O_2$ formation in *Zat12*-transformed lines. The upsurge in ROS is thoroughly related to the damage and stability of membranes and, in so doing, enhanced permeability. MDA stands as a significant marker of cellular damage [38]. Hence, the per-

oxidation of lipids in relation to MDA content was used as a measure to find the integrity of membranes under stress. In this research, temperature stress increased MDA in PBW621, including in other NT wheat cultivars, whereas the smallest increase was observed in *Zat12* transgenics (Figure 3). Together, these outcomes proposed that the intensification in the content of $H_2O_2$ and MDA caused the peroxidation of lipids in NT cultivars under temperature stress. Our results are in accordance with the previous study, where it was been reported that MDA content decreased in plants that are tolerant to high-temperature stress [39].

Enzymes or metabolites augmentation under stress enables plants to overwhelm oxidative stress by creating homeostasis. RT-PCR analysis revealed that the wheat transgenic lines showed a higher expression of the *Zat12* gene under temperature stress (Figure 4), which helps in upregulating the metabolic machinery of antioxidant enzymes. Similarly, Davletova [40] reported that *Zat12* played an important role in stress tolerance and acts as a ROS scavenger. Rizhsky et al. [41] demonstrated that the expression of *Zat12* is unswervingly linked to ROS metabolism in *Arabidopsis thaliana*.

Growth parameter analysis indicated that the treatment of any stress-caused inhibition of both roots and shoot length as, compared to seedlings at control as growth, is susceptible to environmental disturbances; therefore, the growth rate and biomass could be reliable criteria for evaluating plants' ability to withstand stress [42]. Under high-temperature stress conditions, all *Zat12* transgenic lines had significantly increased growth parameters compared to PBW621 (Figures 5 and 6). One transgenic line, Z-8-12 1B, at both conditions (30 °C and 32 °C), was statistically on par with PBW621. Likewise, Davletova [40] observed that, under higher osmotic stress conditions, *Zat12* knock-out plants showed a severe decrease in root and shoot length in comparison to wild-type plants.

The results indicated that overexpressing the *Zat12* gene under stress conferred resistance to heat and cold stress in ZT lines by significantly elevating the antioxidant defense system as compared to NT wheat genotypes, and this may provide a base for the tolerance mechanism of the *Zat12* gene to oxidative stress, which can be considered as a positive feedback mechanism.

In crux, the enhanced expression of the *Zat12* gene under temperature stress, the activation of antioxidant enzymes, and less reduction in growth parameters indicated that *Zat12* ameliorated the temperature stress through convergent influence on osmotic adjustments and augmentation of osmolyte content, and enhanced antioxidant activities, particularly APX; therefore, these results suggest that excess the ROS ($H_2O_2$) which was formed was conceivably detoxified by APX through the Halliwell-Asada cycle and imparts improved defense in transgenics by reducing oxidative stress and lipid peroxidation. Accordingly, it could be concluded that the *Zat12* wheat transgenic lines have an advanced propensity to endure temperature stress, and could possibly be used for wheat cultivation in stress-affected zones to augment yield and food security. Authors of future studies should discuss their results and how they can be interpreted from the perspective of previous studies and of the working hypotheses. The findings and their implications should be discussed in the broadest context possible. Future research directions may also be highlighted.

## 4. Materials and Methods

### 4.1. Material

Seeds of Z-8-12 1A, Z-8-12 1B, Z-8-19, and Z-15-10 [43] were collected along with the selected wheat cultivars (PBW621, PBW550, and HD3086), and *Arabidopsis thaliana* (Col-O) was collected from the Department of Plant Breeding and Genetics, Punjab Agricultural University, Ludhiana. The seeds were surface sterilized by 0.1% $HgCl_2$ for 2 min, cleaned with distilled water multiple times, grown on germination paper in Petri dishes (9 cm) for 24 h at 20 °C, and placed in the dark. Afterward, they were shifted to pots containing vermiculite. The nutrient medium used was Murashige & Skoog media [44]. The trial was separated into three groups. One set of seeds was grown at 20 °C ± 2 °C and was taken

as a control. After 2 weeks, the second group of seeds was transferred to continuous cold stress at $-2\ °C$ and $-4\ °C$ for 7 days, and the third set of seeds was transferred at $30\ °C$ and $32\ °C$ for 7 days. In this way, both cold and heat-stressed seedlings were analyzed at the three-week seedling stage.

### 4.2. Biochemical Analysis

At 21DAG, 15 seedlings were analyzed in individual experiments and recurrent in triplicates. Seedlings (1 g) were homogenized in 50 mM phosphate buffer (pH 6.5) for POD (EC 1.11.1.7) and SOD (EC 1.15.1.1), pH 7.0 for GR (EC 1.6.4.2), and pH 7.5 for GST (2.5.1.18) and APX (1.11.1.1). The homogenate or the mixture was centrifuged for 20 min at $10,000 \times g$ at $4\ °C$ and the clear supernatant was taken for analyzing antioxidative enzymes (APX, SOD, POD, GR, and GST) activity. APX catalyzes the conversion of $H_2O_2$ to $H_2O$ at the cost of ascorbate as an e⁻ acceptor [45]. SOD activity was determined by measuring its capacity to hinder the autooxidation of pyrogallol [46]. POD activity was measured using the method of Shannon et al. [47]. GR was determined by the method of Goyal and Asthir [48]. GST was determined by the measure of its ability to convert hydroperoxide to less toxic dinitrophenyl glutathione using glutathione as substrate by the method of [49]. The content of proline, ascorbate, $H_2O_2$, and MDA was determined by the method of Bates [50], Law et al. [51], Patterson et al. [52], and Heath and Packer [53], correspondingly.

### 4.3. Molecular Analysis

#### 4.3.1. Genomic DNA Extraction

The transgenic status of the *Zat12*-transformed homozygous wheat lines was established by PCR. DNA extracted *Zat12* trangenics leaves, PBW621 (non-transformed), and wild-type *Arabidopsis thaliana* (Col-O) [54]. As *Zat12* gene was isolated from *Arabidopsis thaliana* to make the wheat transformants, it was taken as a positive control. The primers specific to *Zat12* were designed and used to identify the positive transgenic plants against the background of PBW621 using PCR. The primers sequence was *Zat12*-Forward: 5′-TCAGAAGAAAAATGGTTGCGAT-3′, and *Zat12*-Reverse: 5′-GAAAAATTCAAAGAATGAGAG-3′. As the CaMV35S promoter is immediately upstream of the transgene *Zat12*, presence of the promoter indicated the presence of a gene of interest. Therefore, the forward primer of CaMV 35S and the reverse primer of the *Zat12* gene were used.

#### 4.3.2. RT-PCR Analysis

In order to analyze the transgene expression in wheat transgenics, RT-PCR was performed. Whole RNA was isolated from wheat transgenics and PBW621 at 5, 10, and 14 days after the germination stage using the phenol/SDS method [55]. cDNA was created with the help of the Prime Script RT reagent Kit (Takara, Dalian, China). Actin was designated as a housekeeping gene and primers were designed for both *Zat12* and housekeeping genes. The specific primers of housekeeping genes were Actin-Forward: 5′-CCCAAGGCTAACAGGGAGAA-3′, and Actin-Reverse: 5′-GATACCTGTTGTGCGTCCAC-3′. The reaction conditions were standardized as preliminary denaturation at $94\ °C$ for 4 min, denaturation at $95\ °C$ for 1 min, repeated for 35 cycles, annealing at $55\ °C$ for the 30 s followed by extension at $72\ °C$ for 1 min, and then final extension at the same temperature for 10 min.

### 4.4. Analysis of Growth Parameters

The high temperature was used for creating heat stress as a convenient method for studying stress tolerance in transformed lines. The seeds were sown in three sets—one as control and the other two demonstrated at $30\ °C$ and $32\ °C$. They were kept in a growth chamber preserved at $20\ °C \pm 2\ °C$, 8/16 h dark/light for the first seven days. From 8DAG, two sets were maintained at $30\ °C$ and $32\ °C$, respectively, for seven days, whereas the third set was maintained as control under normal conditions for the next seven days. Thus,

after 14DAG, they were analyzed for the growth parameters including the length of the root and shoot and the fresh and dry weight of the root and shoot.

*4.5. Statistical Analysis*

Data were evaluated using two-way analysis of variance (ANOVA) in a two-factor CRD design repeated in triplicate using OPSTAT [56] in accordance with mean values associated with critical difference values (CD) that are significant at a 5% significance level. Data represent triplicate mean values; error bars signify the standard deviation of triplicates. SPSS 13.0 Software was used for post hoc analysis through Tukey's multiple comparison tests and different letters specify significant differences amongst the wheat transgenics/cultivars and temperatures ($p < 0.05$).

**Author Contributions:** Conceptualization, M.K. and B.A.; Data curation, B.A. and A.C.; Formal analysis, M.K. and A.C.; Funding acquisition, B.A.; Investigation, B.A.; Methodology, M.K.; Project administration, B.A.; Resources, B.A. and R.K.; Software, M.K.; Supervision, B.A.; Validation, B.A. and R.K.; Visualization, B.A.; Writing—original draft, M.K.; Writing—review & editing, M.K., B.A. and A.C. All authors have read and agreed to the published version of the manuscript.

**Funding:** This research received no external funding.

**Data Availability Statement:** The data presented in this study are available on request from the corresponding author.

**Acknowledgments:** We generously thank Navtej Singh Bains for his conceptualization and guidance throughout the work.

**Conflicts of Interest:** The authors declare no conflict of interest.

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
