# Peer review of "Zat12 Gene Ameliorates Temperature Stress in Wheat Transgenics by Modulating the Antioxidant Defense System"

_stresses, doi:10.3390/stresses3010023_

Round 1
Reviewer 1 Report
Temperature is one of the major impact factor that influencing crop activities. The current study could be considered for publication with minor revisions unless other reviewers are not against. Some comments listed below should be concerned.
a. It seems that some error bars in Fig. 4 are much wider, please check and include some necessary descriptions or analysis about this condition.
b. The current version only mentioned the procedures of the statistical analysis. It is suggested that the replicated times of the sample tests should also be indicated so that readers could understand the replications of the experiment.
c. Please check the journal guideline if all references require the doi link. If so, please add some missing ones.
d. If possible, some tables with data could also let people understand the research much better.
Reviewer 2 Report
1) Your should decipher the abbreviations DAG (abstract, figure 1), APX, SOD, POD, GST, GR (page 2, figure 1), MDA (figure 3), FW, DW (page 6) for the first time.
2) You should indicate significant differences by different letters (figures 4, 5).
3) In Discussion add references to pictures after the sentences concerning your results. For example, "An advanced APX activity.. (Figure 1)", "The high proline content in transformed wheat lines... (Figure 3)".
4) You should check the design and English in 2-4 paragraphs of the Discussion.
5) If possible, add information and reference about the influence of enhanced Zat12 gene expression on plant yield/productivity/crop quality.
6) If possible, change old references number 25, 26 to newer ones.
